# LEARNING MONOTONIC ALIGNMENTS WITH SOURCE-AWARE GMM ATTENTION

## ABSTRACT

Transformers with soft attention have been widely adopted in various sequence-to-sequence (Seq2Seq) tasks. Whereas soft attention is effective for learning semantic similarities between queries and keys based on their contents, it does not explicitly model the order of elements in sequences which is crucial for monotonic Seq2Seq tasks. Learning monotonic alignments between input and output sequences may be beneficial for long-form and online inference applications that are still challenging for the conventional soft attention algorithm. Herein, we focus on monotonic Seq2Seq tasks and propose a source-aware Gaussian mixture model attention in which the attention scores are monotonically calculated considering both the content and order of the source sequence. We experimentally demonstrate that the proposed attention mechanism improved the performance on the online and long-form speech recognition problems without performance degradation in offline in-distribution speech recognition.

## 1 INTRODUCTION

In recent years, transformer models with soft attention have been widely adopted in various sequence generation tasks (Raffel et al., 2019; Vaswani et al., 2017; Parmar et al., 2018; Karita et al., 2019). Soft attention does not explicitly model the order of elements in a sequence and attends all encoder outputs for each decoder step. However, the order of elements is crucial for understanding monotonic sequence-to-sequence (Seq2Seq) tasks, such as automatic speech recognition (ASR), video analysis, and lip reading. Learning monotonic alignments enables the model to attend to a subset of the encoder output without performance degradation in these tasks. In comparison, soft attention is not suitable for streaming inference applications because the softmax operation needs to wait until all encoder outputs are produced. Figure 1 (b) shows the attention plot for soft-attention. Soft attention learns the alignments between queries and keys based on their similarities; it requires all encoder tokens prior to the attention score calculation. Furthermore, soft attention cannot easily decode long-form sequences that are not considered in the training corpus.

The Gaussian Mixture Model (GMM) attention Graves (2013); Battenberg et al. (2020); Chiu et al. (2019) have been proposed for learning the monotonic mapping between encoder and decoder states for long-form sequence generation. The GMM attention is a pure location-aware algorithm in which encoder contents are not considered during attention score calculation. However, each element in the encoder output sequence contains different amounts of information and should be attended considering their contents. In figure 1 (c), the GMM attention fails to learn the detailed alignments and attends to many tokens simultaneously.

In this study, we adopted the GMM attention mechanism to the modern transformer structure and proposed the Source-Aware Gaussian Mixture Model (SAGMM) attention which considers both contents and orders of source sequences. Each component in the SAGMM is multi-modal and discards non-informative tokens in the attention window. For online inference, we propose a truncated SAGMM (SAGMM-tr) that discards the long-tail of the attention score in the SAGMM. To the best of our knowledge, this is the first

attempt to adopt a GMM-based attention to online sequence generation tasks. Learning accurate monotonic alignments enables the SAGMM-tr to attend to a relevant subset of sequences for each decoder step and improves the performance of the model in terms of streaming and long-form sequence generation tasks. Figure 1 (d) shows the monotonic alignments learned by the SAGMM-tr, enabling online inference. Experiments involving streaming and long-form ASR showed substantial performance improvements compared with conventional algorithms without performance degradation in offline in-distribution ASR. Furthermore, we tested the SAGMM-tr in a machine translation task and demonstrated the performance of the proposed algorithm in non-monotonic tasks.

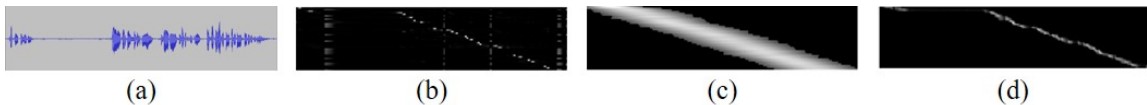

Figure 1: Input speech waveform (a) and attention plots between speech waveform and transcription learned by (b) soft attention, (c) GMM attention, and (d) SAGMM-tr attention.

## 2 SOURCE-AWARE GMM ATTENTION

### 2.1 SOFT ATTENTION

Herein, we abbreviate the head index $h$ during attention score calculation for simplicity. In dot-product multi-head soft attention Vaswani et al. (2017) without relative positional encoding , the attention score from soft attention $\alpha_{Soft}$ is derived from the query matrix $Q \in \mathbb{R}^{I \times d}$ and key matrix $K \in \mathbb{R}^{J \times d}$ as follows:

$$\alpha_{Soft} = \text{softmax}\left(QK^{\mathsf{T}}/\sqrt{d}\right) \tag{1}$$

where $d, I$, and $J$ are the feature dimension, decoder and encoder sequence length, respectively. The attention context matrix from the $h$-th head $\mathrm{H}^h$ and the multi-head output $\mathrm{M}$ are expressed as

$$\mathrm{H}^h = \alpha_{Soft}^h V^h \tag{2}$$

$$\mathrm{M} = \text{concat}\left[\mathrm{H}^1; ...; \mathrm{H}^{n_h}\right] W_O \tag{3}$$

where $\alpha_{Soft}^h$ denotes the $\alpha_{Soft}$ for the $h$-th head, $V^h \in \mathbb{R}^{J \times d}$ the value matrix for the $h$-th head, and $n_h$ the number of heads. In this study, we adopted relative positional encoding Shaw et al. (2018); Dai et al. (2019) which provides a stronger baseline for long-form sequence generation for self-attention layers.

### 2.2 GMM ATTENTION

The previous studies regarding GMM attention Battenberg et al. (2020); Chiu et al. (2019) were based on early content-based attention (Cho et al., 2014). Li et al. (2020) adopted the GMM attention to the transformer framework, but did not provide detailed descriptions. Herein, we adopt v2 model in Battenberg et al. (2020) which improved the performance of the original GMM attention mechanism Graves (2013).

We define the GMM attention as a variant of multi-head attention by considering a Gaussian distribution component as an attention score of single-head in a multi-head mechanism. In the study by Battenberg et al. (2020), the value matrix was shared for all Gaussian components, whereas multi-head value matrices were multiplied with the probability from corresponding components in this study to attend to information from different representation subspaces (Vaswani et al., 2017). Hence, the multi-head GMM attention introduced here is a more generalized algorithm compared with the early GMM attention.

Let us denote the $i$-th row of $Q$ as $Q_i \in \mathbb{R}^{1 \times d}$. The normal distribution parameters for the $i$-th step are expressed

$$\Delta_i,\ \sigma_i,\ \phi_i = \zeta\left(Q_i W_\Delta\right),\ \zeta\left(Q_i W_\sigma\right),\ Q_i W_\phi \tag{4}$$

$$\mu_i = \Delta_i + \mu_{i-1} \tag{5}$$

where $\zeta(x)$ is a softplus function of $x$; $W_\Delta \in \mathbb{R}^{d \times 1}$, $W_\sigma \in \mathbb{R}^{d \times 1}$, and $W_\phi \in \mathbb{R}^{d \times 1}$. The softplus function was adopted, similar to the study of Battenberg et al. (2020) in which softplus activation demonstrated better performances than the exponential operation. A mixture component of the GMM attention, from the $i$-th decoder step to the $j$-th encoder token, is derived from the $\alpha_{GMM\|i,j}$ is defined as follows:

$$N(j, \mu_i, \sigma_i) = \frac{1}{\sqrt{2\pi\sigma_i}} exp\left(-\frac{(j-\mu_i)^2}{2\sigma_i}\right) \tag{6}$$

$$\alpha_{GMM\|i,j} = N(j, \mu_i, \sigma_i) \tag{7}$$

$$\mathrm{H}_i^h = \mathrm{softmax}_h\left(\phi_i^h\right) \sum_j \alpha_{GMM\|i,j}^h V_j^h. \tag{8}$$

where $\mathrm{softmax}_h$ denotes the softmax function over heads.

The conventional GMM attention mechanism is analogous to integral of source sequences with a uniform axis spacing as shown in Figure 2 (a). In this figure, each rectangle denotes the attention score with specified Gaussian component parameters. The GMM attention assumes that each encoder output is equally important. However, this assumption is not satisfied for many input modalities, e.g. speech and videos from real environments. Moreover, the number of modes in the GMM attention is limited by the number of mixture components. To learn robust alignments for monotonic Seq2Seq tasks, we propose the SAGMM which considers both contents and locations for the attention mechanism.

## 2.3 SOURCE-AWARE GMM ATTENTION

Figure 2 (b) shows the scheme of the proposed SAGMM attention. Compared with the GMM attention, the SAGMM is analogous to an integral of normal distribution with non-uniform spacing based on encoder contents. In the figure 2 (b), the width of rectangle $\delta_j$ controls the model to selectively attend to the informative tokens during the attention score calculation. This content-aware property of the SAGMM enables the model to learn stable monotonic alignments from the training corpus. Furthermore, SAGMM can easily discard non-informative tokens and aggregate information distributed over several remote tokens.

In the SAGMM, normal distribution parameter $\Delta_i, \sigma_i, \phi_i$, and $\mu_i$ are derived from equation 4 and equation 5. To encode the contents of the source sequences, the weight for each encoder output $\delta_j$ is provided from the $j$-th row of the key matrix $K$ as follows:

$$\delta_j = \mathrm{sigmoid}\left(K_j W_\delta\right). \tag{9}$$

where $W_\delta \in \mathbb{R}^{d \times 1}$ and the sigmoid function are introduced to smoothly bound the maximum weight $\delta_j$ similar to Dong & Xu (2020).

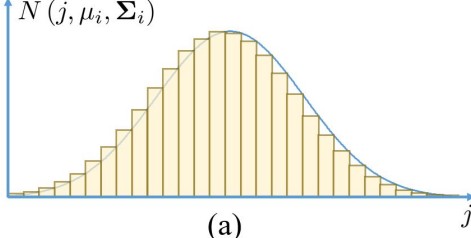
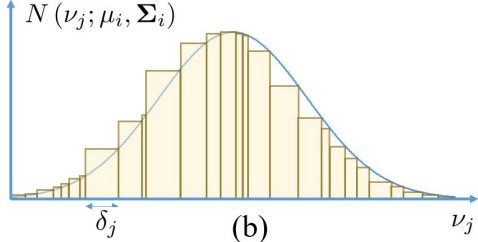

Figure 2: Schemes of (a) conventional GMM and (b) SAGMM attention score calculation which are similar to the integral of the normal distribution on uniform and non-uniform axis spacing, respectively.

Subsequently, the probability of normal distribution $N(\nu_j; \mu_i, \sigma_i)$ is calculated from its cumulative sum $\nu_j$ expressed as

$$\nu_j = \delta_j + \nu_{j-1} \tag{10}$$

$$N(\nu_j; \mu_i, \sigma_i) = \frac{1}{\sqrt{2\pi\sigma_i}} exp\left(-\frac{(\nu_j - \mu_i)^2}{2\sigma_i}\right) \tag{11}$$

Finally, the SAGMM attention score $\alpha_{SAGMM}$ is defined as follows:

$$\alpha_{SAGMM\|_{i,j}} = \delta_j \, N(\nu_j; \mu_i, \sigma_i) \tag{12}$$

where $\delta_j$ where $\delta_j$ is multiplied to describe uneven step sizes in summation.

In the early stage of training, we introduced a length penalty loss to facilitate alignment learning between $\mu$ and $\nu$; it is expressed as

$$L_{length} = \lambda_{length} * \left((\mu_I - \min(I, J))^2 + (\nu_J - \min(I, J))^2\right). \tag{13}$$

with $\lambda_{length} = 0.0005$ where $I$ and $J$ denote the length of the decoder and encoder sequences, respectively. We turned off the $L_{length}$ after 200K of training steps in the experiments. Finally, we modified 5 as $\mu_i = \mu_{i-1} + \min(\max(\Delta_i, 0), 3)$ to scale $\mu_i$ and $\nu_j$ similar to the token indices. This modification facilitates the interpretability of $\mu$ and $\nu$.

The attention score for each encoder token in the SAGMM was determined independently because they do not rely on the softmax operation over the encoder outputs. It is noteworthy that $\sum_j \delta_j \, N(\nu_j; \mu_i, \sigma_i)$ approximates the integral of the Gaussian distribution. Hence, the sum of the attention weights is approximately 1, thereby facilitating numerical stability and learning without using softmax.

## 2.4 SAGMM-TR FOR ONLINE INFERENCE

Since the attention score in SAGMM is generated without softmax normalization over all encoder tokens, we can simply build the attention for streaming inference by cropping the long-tail of the Gaussian distribution. In SAGMM-tr, the normal distribution is truncated to limit the past and future contexts as follows:

$$N_{tr}(\nu_j; \mu_i, \sigma_i) = \begin{cases} \frac{1}{\sqrt{2\pi\sigma_i}} exp\left(-\frac{(\nu_j - \mu_i)^2}{2\sigma_i}\right), & if \; \mu_i - 2\sqrt{\sigma_i} < \nu_j < \mu_i + 2\sqrt{\sigma_i} \\ 0, & else. \end{cases} \tag{14}$$

$$\alpha_{SAGMM\text{-}tr\|_{i,j}} = \delta_j \, N_{tr}(\nu_j; \mu_i, \sigma_i) \tag{15}$$

$$H_i^h = \text{softmax}_h\left(\phi_i^h\right) \sum_{j \in \mu_i - 2\sqrt{\sigma_i} < \nu_j < \mu_i + 2\sqrt{\sigma_i}} \alpha_{SAGMM-tr\|_{i,j}}^h V_j^h. \tag{16}$$

Discarding the tokens with threshold of $2\sqrt{\sigma_i}$ removes approximately 5% of the attention score. In online inference with the SAGMM-tr, $H_i^h$ can be calculated after $\nu_j$ exceeds $\mu_i + 2\sqrt{\sigma_i}$. It is noteworthy that when $\nu_j \geq \mu_i + 2\sqrt{\sigma_i}$ satisfies in for a current token $j = \beta_i$, then the equation satisfies for all $j > \beta_i$ and $H_i^h$ can be emitted without for waiting future contexts.

We started from the SAGMM model and fine-tuned SAGMM-tr until the performance converged. We discovered that the models with the SAGMM-tr demonstrated slightly better performances than the SAGMM; hence, we mainly report the results involving the SAGMM-tr model herein.

For online speech recognition using the SAGMM-tr, we randomly concatenated the **1**-vector after the end of the source sequence with a probability $p_{eos}$ in the training stage and performed training to emit the end-of-sequence ($EOS$) token only with the utterances containing the **1**-vector. The **1**-vector concatenation

suppresses the $EOS$ token in the long silence part. Finally, we built a unidirectional encoder whose maximum latency was similar to those of Zhang et al. (2020); Dong et al. (2019) by adopting a block-wise mask on self-attention layers. A detailed explanation on the block-wise masking is provided in the Appendix. Algorithm 1 in the Appendix shows a pseudo code for the SAGMM-tr attention in inference stage.

The number of tokens required to 16 was determined by model parameters. We trained and tested the SAGMM-tr with a fixed attention window width $c$, to demonstrate the performance in environments with maximum latency constraint. In this version, equation 14 - 16 were modified as follows:

$$\gamma_i = \operatorname*{arg\,max}_{\gamma_{i-1} \le j < J} N\left(\nu_j; \mu_i, \sigma_i\right) \tag{17}$$

$$N_{tr}\left(\nu_j; \mu_i, \sigma_i\right) = \begin{cases} \frac{1}{\sqrt{2\pi}\sigma_i} exp\left(-\frac{(\nu_j - \mu_i)^2}{2\sigma_i}\right), & if \ \gamma - \frac{c}{2} < j < \gamma_i + \frac{c}{2} \\ 0, & else. \end{cases} \tag{18}$$

$$\alpha_{SAGMM\text{-}tr\|_{i,j}} = \delta_j \ N_{tr}\left(\nu_j; \mu_i, \sigma_i\right) \tag{19}$$

$$\mathrm{H}_i^h = \sum_{\gamma_i - \frac{c}{2} < j < \gamma_i + \frac{c}{2}} \alpha_{SAGMM\text{-}tr\|_{i,j}} V_j. \tag{20}$$

In equation 20, we wait $\frac{c}{2}$ additional tokens before producing the output. We compared the performance of the SAGMM-tr with adaptive and fixed window widths through experiments.

## 3 RELATED WORKS

Chiu & Raffel (2018) propose monotonic chunkwise attention (MoChA) based on hard monotonic sampling over attention window (Raffel et al., 2017). The monotonic multi-head attention Ma et al. (2019); Inaguma et al. (2020) and monotonic infinite lookback attention Arivazhagan et al. (2019) are based on the concept similar to that reported by (Chiu & Raffel, 2018). MoChA replaces hard sampling operation to probability distribution over memory in the training stage. By contrast, the SAGMM uses the same operations for training and inference. Furthermore, MoChA relies on a numerically unstable cumulative product that requires additional effort for training (Miao et al., 2019; Raffel et al., 2017). By contrast, the SAGMM adopts a stable cumulative sum to describe monotonic alignments. Finally, MoCha is not effective for long-form speech recognition compared with GMM attention Chiu et al. (2019). Meanwhile, our model outperformed the GMM in the "test-long" set in experiments.

Dong & Xu (2020) introduce the encoder output weight for ASR task. By contrast, our attention mechanism considers the decoder content during attention score calculation. Furthermore, the SAGMM allows the overlap between successive attention windows, facilitating the semantics that span several decoder tokens. Lee et al. (2020) proposed an online softmax-free attention mechanism by aggregating the information of the encoded sequence using an update gate. Similar to soft attention, their approach does not explicitly model monotonic alignments. Furthermore, the update gate for online inference in Lee et al. (2020) is uni-modal.

The location-aware attention Chorowski et al. (2015); Watanabe et al. (2017); Moritz et al. (2019) which introduces previous attention score as additional argument to the current step was proposed to employ positional information in a content-based attention mechanism. The location-aware attention improved the performance of offline inference, whereas we introduced monotonicity in our study for online inference.

Non-attentive neural network-based approaches that do not rely on encoder-decoder attention have been investigated based on Connectionist Temporal Classification (CTC) Graves et al. (2006); Liptchinsky et al. (2017); Li et al. (2019), RNN-transducer (RNN-T) Graves (2012), Transformer-transducer (Transformer-T) Zhang et al. (2020); Yeh et al. (2019), and Imputer Chan et al. (2020). Herein, we focus on the Seq2Seq approach which does not require an assumption regarding sequence lengths and uses the simple beam search algorithm for inference. Block-wise inference has been widely investigated using manually defined decoding

blocks Jaitly et al. (2015); Tsunoo et al. (2020) or joint decoding using a CTC model Moritz et al. (2020). We did not use joint CTC decoding Kim et al. (2016) nor human supervision for the blocked inference.

## 4 EXPERIMENT

We conducted experiments in speech command Warden (2018) and LibriSpeech Panayotov et al. (2015) datasets to compare the performances of our model in standard, online and long-form ASR tasks. We only compared the end-to-end speech recognition model without external language model (LM) rescoring. Furthermore, we performed preliminary experiments on the translation task to demonstrate the performance of the SAGMM-tr on non-monotonic tasks.

### 4.1 EXPERIMENTS ON CONCATENATED SPEECH COMMAND DATASET

First, we performed a speech recognition experiment with limited vocabulary to demonstrate the difficulty of using the typical soft attention algorithm in decoding utterances with unseen sequence lengths. The speech command dataset consists of 1 second long speeches from various speakers uttering single words from a vocabulary of 30 words. We built 100K utterances in the training corpusWarden (2018) by concatenating randomly selected 5-9 utterances in the training data and 500 test utterances by concatenating $\{3, 7, 10, 15, 20\}$ randomly selected words in the test set. We cropped both the start and end of selected utterances randomly from 0.05s to 0.15s before concatenation to prevent the SAGMM from yielding a trivial solution.

We compared the word error rate (WER) of transformers with soft and SAGMM attentions on test sets. As shown in Table 1, the soft attention mechanism tends to memorize the sequence length distribution from the training corpus and fails to decode correctly in the unseen sequence length. Meanwhile, the proposed SAGMM algorithm was robust to sequence length mismatches. Detailed hyperparameters for the experiment, decoded examples for soft attention and the SAGMM are shown in the Appendix.

Table 1: WERs (%) on concatenated speech command dataset with various number of concatenated words.

|  | 3 | 7 | 10 | 15 | 20 |
|---|---|---|---|---|---|
| soft attention | 40.00 | **5.71** | 13.20 | 39.80 | 56.05 |
| SAGMM | **4.67** | 6.29 | **5.50** | **5.40** | **6.45** |

### 4.2 EXPERIMENTS ON LIBRISPEECH DATASET

We conducted experiments on LibriSpeech whose training dataset comprised 960h of read audio books (Panayotov et al., 2015). We validated the performance of the models on development sets and reported the WERs in the tests sets with a fixed beam width of $4$.

First, we compared the performance of the SAGMM-tr with conventional algorithms in offline speech recognition. We trained the multi-head transformers similarly to previous experiments. The encoder comprised 10 layers of self-attention blocks, with 768 hidden nodes and 3,072 filters. The decoder was constructed by stacking 4 self-attention blocks with the same node and filter sizes. The encoder and decoder self-attention employed soft-attention with relative positional encoding (Dai et al., 2019). We trained and tested models that employed soft attention, GMM attention, and SAGMM-tr attention mechanisms as the encoder-decoder attention. We adopted the auxiliary CTC loss on the encoder output with $\lambda_{ctc} = 0.1$ for a fair comparison with the previous attention-based models Lee et al. (2020); Dong & Xu (2020); Kim et al. (2019). The WER of the SAGMM-tr model without CTC loss was 3.84% on test-clean dataset. Detailed hyperparameters used in the experiment and the performance of SAGMM-tr without CTC loss are shown in the Appendix.

Table 2 shows the performance of various non-streamable models. In the non-streamable models, our baseline transformer demonstrated performances similar to those of previous studies considering the number of parameters. Furthermore, we compared the performances of GMM and SAGMM-tr models with a bidirectional encoder to those of other streamable decoder algorithms. The performance of the SAGMM-tr

Table 2: WERs (%) of offline speech recognition models without LM on LibriSpeech (with number of parameters when available).

| | | test-clean | test-other |
|---|---|---|---|
| *Non-streamable* | LAS (Zeyer et al., 2018) | 4.70 | 15.20 |
| | Location-aware attention Moritz et al. (2019) | 6.1 | 20.0 |
| | LAS (180M) (Park et al., 2019) | 3.4 | 8.3 |
| | LAS (361M) (Park et al., 2019) | **2.8** | **6.8** |
| | Transformer (270M) (Synnaeve et al., 2020) | 2.89 | 6.98 |
| *Streamable decoders,* | RNN-T (130M) (Zhang et al., 2020) | 3.2 | 7.8 |
| *bi-directional encoders* | Transformer-T (45.7M) (Yeh et al., 2019) | 6.08 | 13.89 |
| | Transformer-T (139M) (Zhang et al., 2020) | **2.4** | **5.6** |
| | DecGRC (Lee et al., 2020) | 4.83 | 14.90 |
| | Mocha (Lee et al., 2020) | 4.95 | 15.32 |
| | CIF (Dong & Xu, 2020) | 3.41 | 12.62 |
| *Transformer(Ours)* | Soft attention (121M) | **3.20** | **7.96** |
| | GMM (121M) (Ours) | 3.71 | 8.52 |
| | SAGMM-tr (121M) (Ours) | **3.21** | **7.99** |

attention was consistent with that of our transformer model, whereas it was better compared with those of other streamable Seq2Seq models. The results showed that the SAGMM-tr successfully described the monotonic alignments between speech and transcription without attending to all encoder outputs.

Next, we investigated the performance of the SAGMM-tr in an online ASR task. We trained single-head and multi-head SAGMM-tr models using the block-wise masked encoder. Additionally, the truncated GMM model was tested, but discarded in experiments because the model attended to the future context as shown in A.4 and failed to learn the correct alignments. In the multi-head SAGMM-tr, several heads in the first decoder layer were not trained and softmax($\phi_i^h$) approached zero. Therefore, we pruned these heads after the performance on the development set had converged and fine-tuned for 30K additional steps. We believe that training and head pruning with $\phi_i^h$ can prevent over-fitting when the desired monotonic alignments do not require many heads, particularly in the first decoder layer.

Table 3 shows the performances of models based on various online inference approaches. As shown, the single-head SAGMM-tr outperformed other CTC and attention-based algorithms. The multi-head mechanism further improved the performance of SAGMM-tr. Adopting unidireciotnal encoder with block-wise mask increased the WER on the test-clean set by $0.31\%$. Our model demonstrated slightly worse performance than state-of-the-art algorithms with transformer-T (Zhang et al., 2020). Subword regularization Kudo (2018); Hannun et al. (2019) and sequence-level loss Sabour et al. (2019); Prabhavalkar et al. (2018) may enable the SAGMM-tr to achieve performances similar to the best transducer models. Furthermore, the SAGMM-tr can be adopted in various monotonic Seq2Seq tasks easily; this is because it can be inferred via a simple beam search without probability marginalization in transducers and does not require the assumption that the target should be equal or shorter than the source.

Figure 3 shows the attention plots of the single-head SAGMM-tr which successfully captures alignments between speech and transcription for an utterance with a long silence. Furthermore, Figure 1 shows an

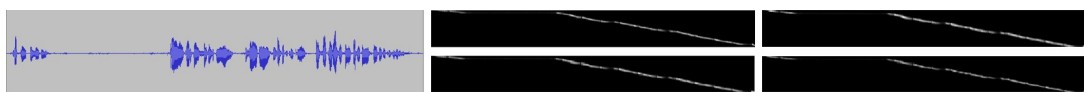

Figure 3: Attention plots for single-head SAGMM-tr model with a speech utterance with a long silence. Each subfigure shows attention plot from each encoder-decoder attention layer in decoder with four blocks.

Table 3: WERs (%) of online speech recognition models without LM on LibriSpeech (with number of parameters when available).

| | | test-clean | test-other |
|---|---|---|---|
| *CTC variants* | Gated ConvNet (Liptchinsky et al., 2017) | 6.7 | - |
| | CTC (Chan et al., 2020) | 4.6 | 13.0 |
| | Imputer(DP)(Chan et al., 2020) | 4.0 | **11.1** |
| | Jasper (333M) (Li et al., 2019) | **3.86** | 11.95 |
| *Transducers* | Transformer-T (45.7M) (Yeh et al., 2019) | 6.47 | 15.79 |
| | ConvT-T (67M) (Huang et al., 2020) | 3.5 | 8.3 |
| | Transformer-T (139M) (Zhang et al., 2020) | **3.0** | **7.7** |
| *Streamable Seq2Seq* | Triggered Attention (Moritz et al., 2019) | 7.4 | 19.2 |
| | Mocha (Kim et al., 2019) | 6.30 | 18.41 |
| | Mocha (Lee et al., 2020) | 5.15 | 16.45 |
| | DecGRC (Lee et al., 2020) | 5.87 | 17.04 |
| | CIF (Dong & Xu, 2020) | 3.96 | 11.19 |
| | SAGMM-tr (121M) (single-head) | 3.73 | 9.92 |
| | SAGMM-tr (121M) (multi-head) | **3.52** | **9.29** |

Table 4: Experimental results of (a) SAGMM-tr with fixed window width and (b) long-form speech recognition on "test-long" set. (sh) and (mh) denote single-head and multi-head attention, respectively.

(a)

| | test-clean | test-other |
|---|---|---|
| SAGMM-tr (sh) | 3.73 | 9.92 |
| SAGMM-tr (mh) | **3.52** | **9.29** |
| SAGMM-tr (sh,fixed) | 3.67 | 9.76 |
| SAGMM-tr (mh,fixed) | 4.45 | 10.34 |
| +fine-tuning | 3.60 | 9.42 |

(b)

| | test-clean | test-long |
|---|---|---|
| Soft attention | **3.20** | 32.77 |
| GMM | 3.71 | 24.71 |
| SAGMM-tr (sh) | 3.73 | **5.60** |
| SAGMM-tr(mh) | 3.52 | 8.13 |

attention alignments from multi-head soft attention, truncated GMM attention and SAGMM-tr attentions(the full plots are shown in Appendix A.4). In soft attention, several heads attend to the speech absence frames constantly for every decoder step. These heads might facilitate in the utterance-level normalization on the attention context vectors. In future studied, the role of silence-attending heads in soft attention should be investigated and introduced to the SAGMM-tr attention. The truncated GMM model failed to learn the alignment and attended to a wide range of encoder tokens. Finally, the SAGMM-tr attention learned the monotonic alignments successfully. The SAGMM-tr attention plots in the lower layers were more blurred compared with the higher layers. We suppose that the lower layers found the relevant subsets in the sequence and the higher layers utilized the context vectors from the lower encoder-decoder attentions and relatively more focused, partly resembling the behavior of soft attention.

The multi-head SAGMM-tr model showed more blurred alignments compared with the single-head model. Our hypothesis for this phenomenon is that several heads attended larger windows to provide context information. The fixed attention window width in equation 17 - 20 or regularization on $\sigma_i$ can minimize the maximum latency of online inference. In this study, we tested the fixed attention window width. We began with the SAGMM-tr models trained for the online ASR experiment and measured the WERs before and after fine-tuning with a constant window width ($c = 15$ frames). From the results in Table 4 (a), the SAGMM-tr attention mechanism can be adopted to the models with a fixed attention window width.

Finally, we concatenated utterances from the same speaker in the "test-clean" set to build a "test-long" set for the long-form speech recognition. The average length of utterances in the test-long set was 54s. In the SAGMM-tr model, the beam search algorithm was slightly modified from Algorithm 1 to suppress the $EOS$ token until the attention window encompassed $\nu_{h,J}$. The performance of the GMM attention did not improved by the same modification since GMM attention attends to $\nu_{h,J}$ prior to the end of a sentence.

As shown in Table 4 (b), the SAGMM-tr outperformed the conventional soft and GMM attention models in long-form speech recognition. Comparing the SAGMM-tr models, the multi-head SAGMM-tr showed worse performance than the single-head model. The performance degradation of the multi-head mechanism on the "test-long" set might arise from accumulated attention windows range mismatches. Attention window ranges in multi-headed models should be investigated in future studies.

## 4.3 EXPERIMENTS ON NON-MONOTONIC TASK

The proposed algorithm focuses on strict monotonic tasks that do not allow reordering. However, the self-attention layers in the encoder can reorder the semantics in input sequences before monotonic alignments from the SAGMM-tr. We conducted machine translation experiments on the WMT EN-DE environment to demonstrate the performance of the SAGMM-tr attention in non-monotonic tasks. Whereas the machine translation is a non-monotonic task, various approaches have been proposed to improve the performance of online inference and simultaneous translation (Arivazhagan et al., 2019; Elbayad et al., 2020). In this study, we conduct simple experiments to figure out if the GMM attention can manage non-monotonic tasks.

We trained the encoder-decoder model similar to the transformer-base in Vaswani et al. (2017) except that the head size was $4$ in the encoder-decoder and decoder self-attention. We trained a model with 4.5M pairs of sentences and measured the BLEU score on the newstest2013 set after performing 100K of training steps. For the uni-directional encoder, a block-wise mask with $M = 5$ was applied to the SAGMM-tr model with a bi-directional encoder and then fine-tuned for 40K additional steps.

Table 5 shows the performances of soft and SAGMM-tr attention models on the newstest2013 set. As expected, the performance of the SAGMM-tr was not consistent with that of conventional transformer models with soft attention. Furthermore, the uni-directional encoder deteriorated the performance of the SAGMM-tr, similar to the speech recognition experiments. Because the SAGMM-tr does not consider local reordering, it is difficult for the proposed algorithm to learn locally non-monotonic functions. In our future studies, we will improve the performance of SAGMM-tr for local reordering tasks by positive constraint relaxation on $\Delta_i$.

Table 5: BLEU scores on newstest2013 set for WMT EN-DE environment.

|  | EN-DE |
| --- | --- |
| Transformer (Vaswani et al., 2017) | 25.8 |
| Soft attention (Ours) | 25.7 |
| SAGMM-tr (bi-enc) | 24.0 |
| SAGMM-tr (uni-enc) | 23.7 |

## 5 CONCLUSION

We proposed the SAGMM, which attends the subset of source sequences according to the normal distribution on a weighted encoder output axis, considering both contents and order of elements in sequences. Furthermore, we proposed the SAGMM-tr for online/real-time inference applications. Based on results on various speech recognition tasks, it was discovered that the proposed attention mechanism can learn monotonic alignments between source and target sequences without human supervision. The performance of a transformer with the SAGMM-tr improved for online and long-form speech recognition without performance degradation in the standard offline ASR task. In future studies, we plan to adopt SAGMM-based attention mechanisms for natural language processing tasks that allow local reordering during sequence generation. We are also interested in latency minimization for online inference by controlling $\mu$ and $\sigma$ in both the training and inference stages.

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

# A APPENDIX

## A.1 BLOCK-WISE MASK FOR ENCODER SELF-ATTENTION LAYER

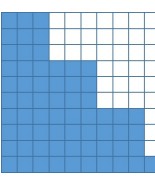

Figure 4: Encoder self-attention mask between query (vertical axis) and key (horizontal axis) in $M = 3$ case.

The block-wise uni-directional mask was adopted for encoder self-attention layers to block future context more than $M$ tokens. 4 shows the block-wise masking with $M = 3$ case. In the experiment with SAGMM-tr, we chose $M = 30$ whose gmaximum latency is comparable to other studies with uni-directional encoders (Zhang et al., 2020; Dong et al., 2019). The blcok-wise masking is different from lookahead operation since encoder outputs from the block-wise masking are not guaranteed to look future $M$ token depends on their position in block while the lookahead operation provides maximum future context for all encoder outputs.

## A.2 PSEUDO CODE FOR ONLINE INFERENCE WITH SAGMM-TR ATTENTION

---
**Algorithm 1:** Decoding process of SAGMM-tr attention mechanism for a $h$-th head.

---
**Data:** encoder key and value vectors $K_j^h, V_j^h$ for $j \in \{1, 2, ...J\}$, $i = 1$, $\mu_0^h = \nu_0^h = 0$, $y_0 = SOS$
**while** $y_{i-1} \neq EOS$ **do**
  $\quad Q_i^h = \text{feedforward}(y_{i-1})$                                 `/* Lower decoder layers */`
  $\quad \Delta_i^h, \sigma_i^h, \phi_i^h = \text{GMMparam}(Q_i^h), \quad \mu_i^h = \mu_{i-1}^h + \min(\max(\Delta_i^h, 0), 3)$    `/* equation 4 */`
  $\quad pos_s^h, = \mu_i^h - 2\sqrt{\sigma_i^h}, \quad pos_e^h = \mu_i^h + 2\sqrt{\sigma_i^h}, Attend^h = \mathbf{0}$
  $\quad$**for** $j = 1$ *to* $J$ **do**
  $\quad\quad \delta_j^h = \text{sigmoid}(K_j^h W_\delta^h), \quad \nu_j^h = \nu_{j-1}^h + \delta_j^h$
  $\quad\quad$**if** $\nu_j^h > pos_e^h$ **then**
  $\quad\quad\quad$**Break**
  $\quad\quad$**else if** $\nu_j^h > pos_s^h$ **then**
  $\quad\quad\quad \alpha_{SAGMM\text{-}tr\|i,j}^h = \delta_j^h N_{tr}(\nu_j^h; \mu_i^h, \sigma_i^h)$                   `/* equation 14, 15 */`
  $\quad\quad\quad Attend^h = Attend^h + \alpha_{SAGMM\text{-}tr\|i,j}^h V_j^h$
  $\quad y_i = \text{Output}(\text{Decoder}(\text{softmax}_h(\phi_i^h) Attend^h)), \quad i = i + 1$

---

## A.3 EXPERIMENT CONFIGURATIONS

For all speech recognition experiments, the 80-dimensional log-Mel filterbank features were extracted with a 25ms window and shifted every 10ms. Each input vector stacked 3 log-Mel feature vectors, downsampled to 30ms frame rate.

In the speech command dataset experiment, We trained the multi-headed transformer Vaswani et al. (2017); Dong et al. (2018) with soft attention and SAGMM attention for encoder-decoder alignments. We adopt the

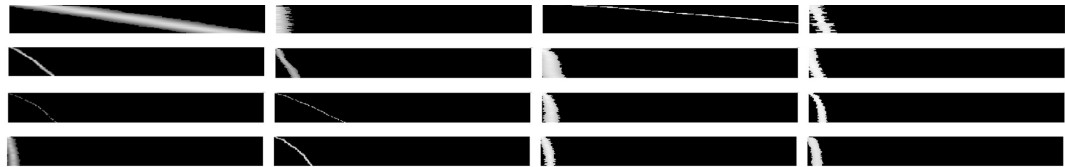

Figure 5: Attention alignments plots from multi-head truncated GMM model.

relative positional encoding for both the encoder and decoder self-attention (Dai et al., 2019). The encoder consisted of 6 layers of self-attention block with 512 nodes and 2,048 filter size, respectively. The decoder was constructed by stacking 3 self-attention block with the same node and filter sizes. The number of heads for encoder self attention, encoder-decoder attention, and decoder self attention were $\{8, 4, 4\}$, respectively. We employed label smoothing of value $\epsilon_{ls} = 0.15$, and applied parallel scheduled sampling with probability 0.2 after 100k (Duckworth et al., 2019). We used the Adam optimizer with $\beta_1 = 0.9$, $\beta_2 = 0.98$ and $\epsilon = 10^{-9}$). The learning rate was linearly increased to 0.1 until 16k steps and set constant in following training steps. We used the tensorflow framework and trained the model until 500K step with 4 m40 GPUs. We averaged 5 last checkpoints before the inference. The dropout ratio for attention weights, rectified linear units, output of sub-layers, and neural network input after positional encoding were 0.2, 0.1, 0.2, and 0.2, respectively. The Specaug algorithm Park et al. (2019) was also applied during training. The width of beam was set to 4 in all experiments. The output token segments are grepheme of the reference.

In the LibriSpeech experiment, The learning rate was linearly increased to 0.1 until 16k steps and set constant until the model converges on dev-clean dataset. Then, we lowered learning rate to 0.02 and fine-tuned until the model converges. We used the tensorflow framework and the model is trained from 7 to 10 days on 8 p40 GPUs. We averaged $\{5, 10, 15\}$ last checkpoints before the performance evaluation. The 1K wordpiece vocabularies extracted from training dataset is used for output token segmentation. Other parameters are the same with those in the speech command dataset experiment.

### A.4 ATTENTION PLOT FOR VARIOUS ENCODER-DECODER MODELS

To show the SAGMM-tr is able to learn alignments between encoder and decoders, we plot the attention score for single and multi-head models. We used the utterance "Chapter eleven ¡long pause¿ the morrow brought a very sober looking morning the sun making only a few efforts to appear and Catherine augured from it everything most favourable to her wishes" in LibriSpeech.

The truncated GMM model fails to learn the true alignment attended several distinct windows in 5. Surprisingly, the performance of truncated GMM was 4.03% and 10.16% for test-clean and test-other, respectively. The decent performance of truncated GMM implies that the GMM learns to attend as many frames as possible instead of learning alignments between sequences. Therefore, the truncated GMM attention is not suitable for streaming inference.

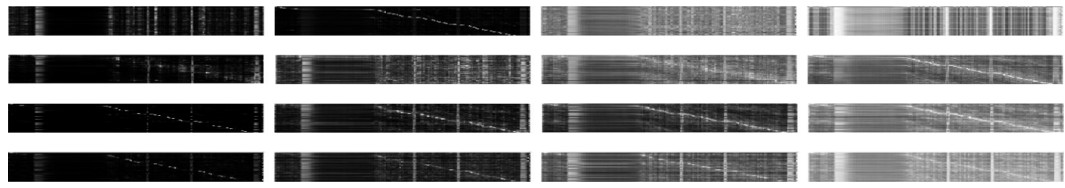

Figure 6: Attention alignments plots from multi-head soft attention model.

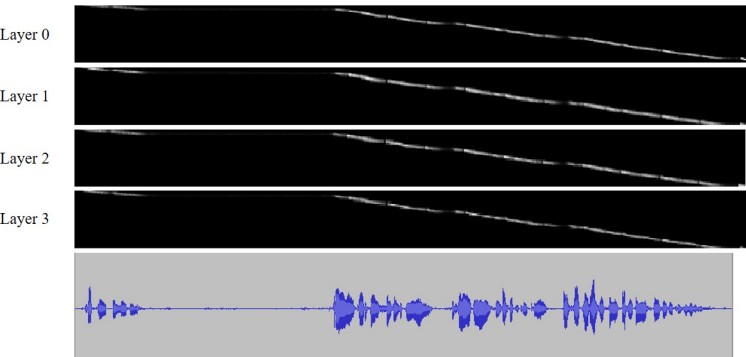

Figure 7: Attention alignments plots for single-head SAGMM-tr for the utterance "Chapter eleven ¡long pause¿ the morrow brought a very sober looking morning the sun making only a few efforts to appear and Catherine augured from it everything most favourable to her wishes" in LibriSpeech. The attention plot in time domain longer than the original wave due to **1**-vector concatenation for $EOS$ 2.4

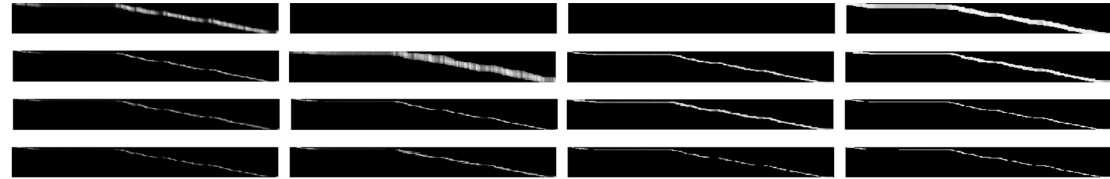

Figure 8: Attention alignments plots from multi-head SAGMM-tr model with adaptive window width. The empty plots denote the pruned head in the first layer

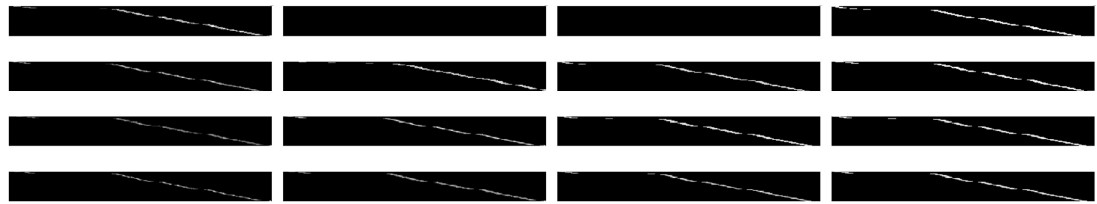

Figure 9: Attention alignments plots from multi-head SAGMM-tr attention model with fixed window width. The empty plots denote the pruned head in the first layer

In the soft attention model, the several heads learns monotonic alignments between inputs and outputs as shown in Figure 6. However, other heads attend to the whole speech presence or absence intervals. We assume these heads help utterance-level normalization. While a few heads show monotonicity, the soft attention is not suitable for online inference due to softmax operation over all frames.

The attention alignment figure for single and multi-headed SAGMM-tr model is shown in Figures 7, 8 and 9. From the figures, both models learn the optimal alignments between input wave and transcription and distinguish the word boundaries without relying on human knowledge. The width of attention windows in multi-head SAGMM-tr with adaptive window width are longer than those in the single-head model. The additional objective function for regularizing $\sigma_i$ would be helpful to reduce the window width in figure 8.

In the multi-head model with fixed window width, the model learns the multi-head monotonic alignment without increasing the window width. Considering the attention window width as a hyperparameter from external knowledge, it is shown that additional information could improve the model latency in SAGMM-tr while the model can also be trained without human knowledge.

### A.5   ABLATION STUDIES

We tested the three ablation experiments on SAGMM-tr attention in Figure 6. First, we tested the model without CTC loss on encoder output. The CTC loss improves the word error rate performances of model similar to previous studies, but the model showed decent performance without the CTC loss. Second, we tested the SAGMM-tr model with uni-directional encoder without initialization from the bi-directional encoder. The result shows slightly worse performance compared to the original SAGMM-tr. Finally, we train the SAGMM model in the training and decoded with the SAGMM-tr decoding scheme without fine-tuning. The performance degradation in 6 shows that the mismatch between training and test stages should be removed by the fine-tuning with truncated normal distribution.

Table 6: The word error rate (%) of ablation experiments for SAGMM-tr attention mechanism in LibriSpeech.

|  | test-clean | test-other |
|---|---|---|
| SAGMM-tr | 3.52 | 9.29 |
| SAGMM-tr w/o CTC loss | 3.84 | 9.86 |
| SAGMM-tr uni-encoder from scratch | 3.69 | 9.79 |
| SAGMM-tr w/o truncated attention fine-tuning | 3.76 | 9.96 |

### A.6   ATTENTION PLOTS FOR SAGMM-TR IN MACHINE TRANSLATION EXPERIMENTS

We plot the first three decoder layers of the SAGMM-tr model with bi-directional encoder for an example utterance "The patient really needs to be made to understand the degree of risk of his cancer, by offering him the options available, not necessarily treating prostate cancers that are not long-term life threatening, and opting instead, in such cases, for active monitoring of the disease." and model output "Der Patient muss wirklich verstanden werden, um den Grad des Risikos seines Krebses zu verstehen, indem er ihm die Möglichkeiten zur Verfügung stellt, nicht notwendigerweise mit Prostatakrebs umzugehen, die nicht langfristiges Leben bedrohen und sich stattdessen in solchen Fällen für eine aktive Überwachung der Krankheit entscheiden.".

We also plot the first three encoder-decoder attention and encoder-self attention layers of the SAGMM-tr model with uni-directional encoder for an example utterance "The patient really needs to be made to understand the degree of risk of his cancer, by offering him the options available, not necessarily treating prostate cancers that are not long-term life threatening, and opting instead, in such cases, for active monitoring of the disease." and model output "Der Patient muss wirklich verstanden werden, um das Risiko seines Krebses zu verstehen, indem er ihm die Möglichkeiten zur Verfügung stellt, nicht notwendigerweise mit Prostatakrebs umzugehen, die nicht langfristiges Leben bedrohen, und stattdessen in solchen Fällen für eine aktive Überwachung der Krankheit zu entscheiden.".

From the figures 10, 11 and 12, the transformers with SAGMM-tr approximates the machine translation task as a strict monotonic task and finds optimal alignments under the assumption. Since the machine translation is not strictly monotonic task, the performance of the SAGMM-tr attention deteriorates compared to the soft attention. However, the transformer with SAGMM-tr attention and uni-directional encoder enables online inference for simultaneous translation.

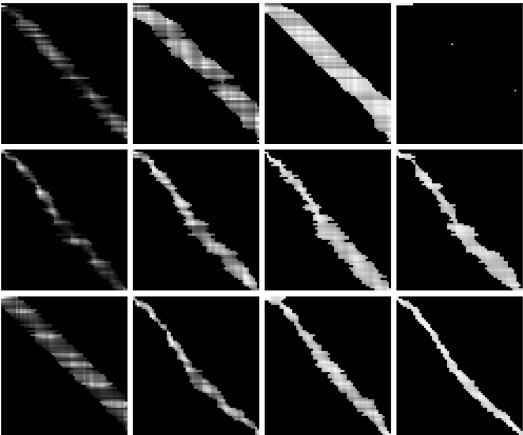

Figure 10: Encoder-decoder attention alignments plots from multi-head SAGMM-tr model with bi-directional encoder in machine translation experiment.

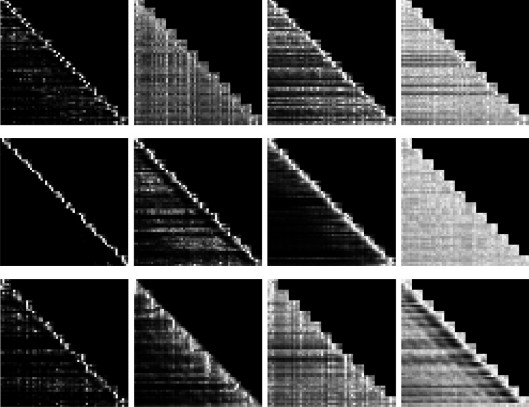

Figure 11: Encoder self-attention alignments plots from multi-head SAGMM-tr model with uni-directional encoder in machine translation experiment.

### A.7 DECODED EXAMPLES OF SOFT ATTENTION AND SAGMM MODELS ON SPEECH COMMAND AND LIBRISPEECH DATASETS

Table 7 and 8 show the decoded examples from the models with unseen sequence lengths. From the tables, soft attention has difficulty to generate the sequences that are significantly shorter or longer than the training corpus distribution. In contrast, the SAGMM-tr is robust to the sequence length mismatch and generates the transcription to arbitrary length.

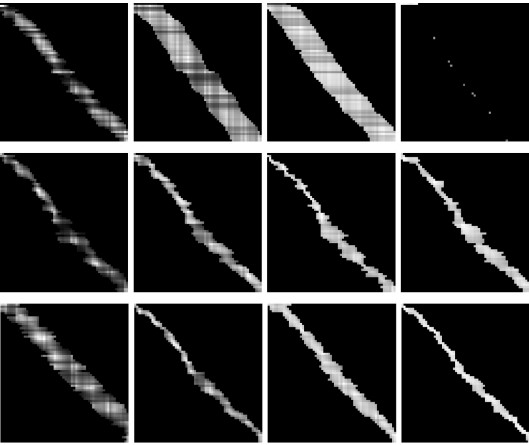

Figure 12: Encoder-decoder attention alignments plots from multi-head SAGMM-tr model with uni-directional encoder in machine translation experiment.

Table 7: The example results for attention-based models in concatenated speech command dataset.

|  | Transcription |
| --- | --- |
| Reference | seven three stop |
| Soft attention | seven three stop stop |
| SAGMM | seven three stop |
| Reference | two right two no cat bird right left tree four tree one wow marvin left |
| Soft attention | two right two no cat bird right left                             marvin left |
| SAGMM | two right two no cat bird right left tree four tree one wow marvin left |

Table 8: The example results for attention-based models in test-clean-long set from LibriSpeech dataset. (sh) and (mh) denote the single-head and multi-head attention, respectively.

| | Transcription |
|---|---|
| Reference | it was idle for him to move himself to be generous towards them to tell himself that if he ever came to their gates stripped of his pride beaten and in beggar's weeds that they would be generous towards him loving him as themselves idle and embittering finally to argue against his own dispassionate certitude that the commandment of love bade us not to love our neighbour as ourselves with the same amount and intensity of love but to love him as ourselves with the same kind of love the phrase and the day and the scene harmonized in a chord words was it their colours they were voyaging across the deserts of the sky a host of nomads on the march voyaging high over ireland westward bound the europe they had come from lay out there beyond the irish sea europe of strange tongues and valleyed and woodbegirt and citadelled and of entrenched and marshalled races |
| Soft attention | it was idle for him to move himself to be generous towards them to tell himself that if he ever came to their gates stripped of his pride beaten and in beggars weeds that they would be generous towards him loving him as themselves to love him as ourselves with the same amount and intensity of love the europe they had come from lay out there beyond the irish sea europe of strange tongues and valleyed and martialed races |
| SAGMM-tr(sh) | it was idle for him to move himself to be generous towards them to tell himself that if he ever came to their gates stripped of his pride beaten and in beggars weeds that they would be generous towards him loving him as themselves idle and embittering finally to argue against his own dispassionate certitude that the commandment of love bade us not to love our neighbour as ourselves with the same amount and intensity of love but to love him as ourselves with the same kind of love the phrase and the day and the scene harmonized in accord words was it their colors they were voyaging across the deserts of the sky a host of no mads on the march voyaging high over ireland westward bound the europe they had come from lay out there beyond the irish sea europe of strange tongues and valleys and wood be girt and set and of intrenched and marshalled races |
| SAGMM-tr(mh) | it was idle for him to move himself to be generous towards them to tell himself that if he ever came to their gates stripped of his pride beaten and in beggar's weeds that they would be generous towards him loving him as themselves idle and embittering finally to argue against his own dispassionate certitude that the command of love bade us not to love our neighbour as ourselves with the same amount and intensity of love but to him as ourselves with the same kind of love the phrase and the day and the scene harmonized in accord words was it their colors they were voyaging across the deserts of the sky a host of no mads on the march voyaging high over ireland westward bound the europe they had come from lay there beyond the irish sea europe of strange tongues and valleyed and wood girt and citadel and of intrenched and marched and races |

