# OpenReview forum: "Learning Monotonic Alignments with Source-Aware GMM Attention"
_ICLR.cc/2021/Conference — Reject_

### Official Review · AnonReviewer2 · 2020-10-28
**Nice idea, but details unclear and presentation needs polish**

**Rating:** 5
**Confidence:** 4

**Review:**

The paper describes a simple extension to the location-only monotonic GMM attention mechanism from Graves (2013), which takes the source/key context into account when computing attention weights.  The proposed method improves ASR performance over model using the baseline GMM attention which does not take source-content into account,  generalizes better to input sequences much longer than those seen during training, while also obtaining competitive performance to other streaming seq2seq ASR models on "matched" test sets.

## Pros:

1. Incorporating source content is an obvious and useful extension to monotonic GMM attention, combining the strengths of content-based approaches such as additive or dot-product attention.

2. It improves performance and generalization while being simpler than existing techniques in the literature (e.g. Mocha, CTC/transducer models which have more complex loss functions).

## Cons:

1. The description of the proposed mechanism is inconsistent with existing literature and is very unclear and confusing in parts.

2. Experiments are somewhat incomplete/missing important comparisons, e.g. comparing baseline GMM attention to the proposed "source-aware" variant in Tables 1,2,5, and comparing to other location-based attention mechanisms, even if non-monotinic, e.g. from http://papers.nips.cc/paper/5847-attention-based-models-for-speech-recognition.pdf

3. Overall the writing/language use could use improvement.

## Detailed comments

At the high level, the idea of incorporating the source keys K into GMM attention is a good one.  The proposed method seems to work, and be  simpler to implement than alternative monotonic alignment mechanisms used in seq2seq ASR models.  However, given the current state of the text, with many confusing details, I feel that the paper is not yet ready for publication without significant revisions.

Many details in the paper, especially Section 2, are unclear:

- Sec 2.2. and throughout the paper:  The described "GMM" and "SA-GMM" attention always use a single component, so don't really count as a Gaussian *mixture*.  Using multiple components would explicitly allow for multimodal attention weights for each output step.  Moreover, since the mixing weights are generally computed independently at each step, using multiple components makes it possible for the base GMM attention to "discard the non-informative tokens".   This mechanism, which would be more precisely called "Gaussian attention", is strictly less flexible than the base GMM attention mechanism that was originally described in Graves, 2013.

- This claim is repeated in paragraph 2 of Sec 3: "uni-modal similar to conventional GMM attention".  When using multiple mixture components, GMM attention is not unimodal.

- Eq 8: The notation here is unclear.  Why is there a softmax over $\psi_i^h$ (a scalar AFAICT)?  Is the softmax computed over all attention heads?   Why is this necessary?  It seems to impair the training of some heads, at least for SAGMM-tr according to paragraph 4 of Sec. 4.2

- Figure 1 is difficult to interpret.  The two plots have difference horizontal axes and therefore don't seem to be directly comparable...  It's not clear what the "key width" in Figure 1b is trying to convey since there is always going to be a single weight per (discrete) encoder step j.

- Sec 2.3.  There is no particular motivation given for the proposed method for integrating source keys.  Why not include $K_j$ in the computation for the standard deviation $\Sigma_i$ as well?  And why is the same weight $\delta_j$ used as a scaling factor (eq (12)) and the mean offset in eq (10)?  These design choices deserve more explanation, and possibly empirical justification.

- Sec 2.4: Is it possible to train SAGMM-tr from scratch?  Or does it need to be first trained using SAGMM and then fine-tuning with truncation enabled?

Experiments:

- Are the different GMM attention variants used encoder self-attention layers as well?  Or does the encoder use conventional "soft attention"?

- As above, it seems unfair not to include any experiments using multiple components when comparing different variants of GMM attention.

- Sec 4.2, Table 1:  Please clarify the differences between the three models labeled (Ours).  Is the difference only in the encoder-decoder attention layer?

English usage.  Just a few examples of grammar errors and unclear text, as there are too many to list.

- page 1, "attend subset of long sequences" is missing a preposition, e.g., "attend to a subset".  It seems that "long sequences" is meant  to refer a single source sequence.

- page 1, "mismatch between attention parameters from decoders and information distribution in encoder outputs".  This sentence is difficult to parse.  What is the mismatch here?   Why would the source encoding and decoder query vector need to "match", especially in a purely location-based attention scheme?

- page 4: "fixed length with hyperparameters"  What are they hyperparameters being referred to here?
- page 5, Sec. 4: "enables early inference".  What does "early inference" mean?
- page 5, Sec. 3: "for the inference" -> "for inference"
- page 5, Sec 4.1:  "1 second speeches" -> "1 second long utterances"
- page 5, Sec 4.1: "from 30 vocabulary" -> "from a vocabulary of 30 words"


Other comments:

- Sec 2.2: Sutskever et al., 2014 did not use content-based attention.

---

> ### Author Response · Authors · 2020-11-18
> **Response to Reviewer #2 (1/2)**
>
> We would like to thank the anonymous reviewers for their valuable feedback, and for the time and effort spent on evaluating our submission. The reviewers raised some interesting points, which we have carefully addressed, and as a result, we believe that the revised draft is significantly improved compared to the first submission.
>
> **Comment** The description of the proposed mechanism is inconsistent with existing literature and is very unclear and confusing in parts.
>
> **Response** Thanks to the reviewers’ comments, we have polished up confusing parts in the paper. We fixed wrong explanation about the multi-modality of conventional GMM. Section 2.4 was revised for readability.
>
> **Comment** Experiments are somewhat incomplete/missing important comparisons, e.g. comparing baseline GMM attention to the proposed "source-aware" variant in Tables 1,2,5, and comparing to other location-based attention mechanisms, even if non-monotonic, e.g. from http://papers.nips.cc/paper/5847-attention-based-models-for-speech-recognition.pdf
>
> **Response** We added mentions for location-aware attention in Section 3 and compared our model with the recent study on location-aware attention [Link]( https://ieeexplore.ieee.org/document/8683510). We fine-tuned our soft-attention model with location-aware term and obtained WERs of 3.08/7.92 for test-clean and test-other, respectively. We did not include the results of our location-aware model in the paper because it was not trained from scratch.
>
> **Comment** Overall the writing/language use could use improvement.
>
> **Response** We are polishing up the paper to improve readability. We will update the proofread in the rebuttal stage.
>
> **Comment** Sec 2.2. and throughout the paper: The described "GMM" and "SA-GMM" attention always use a single component, so don't really count as a Gaussian mixture. Using multiple components would explicitly allow for multimodal attention weights for each output step. Moreover, since the mixing weights are generally computed independently at each step, using multiple components makes it possible for the base GMM attention to "discard the non-informative tokens". This mechanism, which would be more precisely called "Gaussian attention", is strictly less flexible than the base GMM attention mechanism that was originally described in Graves, 2013.
>
> **Response** The GMM attention model in our paper corresponds to the GMM attention from Graves, 2013 and Battenburg et. al, 2020 [Link](https://arxiv.org/abs/1910.10288). We started from the ‘v2’ version model of Battenberg et al., 2020  that improved Graves, 2013. The difference between Battenberg's model and our model is that we adopt the multi-head value matrices. When the value matrices are shared for all heads, our model is similar to Battenberg's model. By introducing multi-head value matrices, our model attends to the different representation subspaces simultaneously. Strictly, our model in session 2.2 is ‘multi-head Gaussian attention’ which is generalized form of the GMM attention. In this paper, we used the same term ‘GMM attention’ since introducing multi-head value matrices is minor update from previous study.
>
> **Comment** This claim is repeated in paragraph 2 of Sec 3: "uni-modal similar to conventional GMM attention". When using multiple mixture components, GMM attention is not unimodal.
>
> **Response** Thanks to the reviewer’s comment, we fixed the wrong description about GMM attention and rewrote as ‘Moreover, the number of modes in GMM attention is limited by the number of mixture components’. in Section 2.2 and ‘Also, the update gate for online inference in Lee et al. (2020) is uni-modal.’ In Section 3.
>
> **Comment** Eq 8: The notation here is unclear. Why is there a softmax over $\phi_{i}^{h}$? Is the softmax computed over all attention heads? Why is this necessary? It seems to impair the training of some heads, at least for SAGMM-tr according to paragraph 4 of Sec. 4.2
>
> **Response** As the reviewer commented, the softmax on $\phi_{i}^{h}$ is conducted over all attention heads. We revised the notation for softmax over attention heads as $softmax_{h}$. We adopt $\phi_{i}^{h}$ to match the model with previous studies on GMM attention. Also,  the redundant heads can be easily pruned by $softmax_{h}$  which helps to prevent over-fitting.
>
> **Comment**  Figure 1 is difficult to interpret. The two plots have difference horizontal axes and therefore don't seem to be directly comparable... It's not clear what the "key width" in Figure 1b is trying to convey since there is always going to be a single weight per (discrete) encoder step j.
>
> **Response** In Figure 2, (figure 1 in the original paper), the difference between (a) and (b) is non-uniform axis spacing which encodes source contents. $\delta_{j}$ represents the amount of information of $j$-th encoder token ands helps the SAGMM attention to attend to informative tokens only.

---

> > ### Author Response · Authors · 2020-11-18
> > **Response to Reviewer #2 (2/2)**
> >
> > **Comment** Sec 2.3. There is no particular motivation given for the proposed method for integrating source keys. Why not include $K_{j}$ in the computation for the standard deviation as well? And why is the same weight $\delta_{j}$ used as a scaling factor (eq (12)) and the mean offset in eq (10)? These design choices deserve more explanation, and possibly empirical justification.
> >
> > **Response** Our motivation for SAGMM is to encode source with $\delta_{j}$ and aggregate information by weighted summation which is analogous to integral of normal distribution on non-uniform axis spacing. The mean offset is directly derived from $\delta_{j}$ to match the integral of normal distribution.  Matching SAGMM to integral of probability density helps the numerical stability of the SAGMM attention mechanism without softmax operation over encoder tokens. We could not include $K_{j}$ into the normal distribution parameters since the Gaussian distribution is shared for all encoder tokens.
> >
> > **Comment** Sec 2.4: Is it possible to train SAGMM-tr from scratch? Or does it need to be first trained using SAGMM and then fine-tuning with truncation enabled?
> >
> > **Response** Training SAGMM-tr from scratch slowed the convergence of model. We recommend to train SAGMM in early stage and fine-tune SAGMM-tr from it.
> >
> > **Experiments**
> > **Comment** Are the different GMM attention variants used encoder self-attention layers as well? Or does the encoder use conventional "soft attention"?
> >
> > **Response** Our models adopt the same the encoder and decoder self-attention layers based on the soft-attention with relative positional encoding. We added the sentence ' The encoder and decoder self-attention employed soft-attention with relative positional encoding' in Sec 4.2 for readability.
> >
> > **Comment** As above, it seems unfair not to include any experiments using multiple components when comparing different variants of GMM attention.
> >
> > **Response** We compared our model with location-aware attention in Table 2.
> >
> > **Comment** Sec 4.2, Table 1: Please clarify the differences between the three models labeled (Ours). Is the difference only in the encoder-decoder attention layer?
> >
> > **Response** The difference between three models is the encoder-decoder attention layer. We added the explanation as ‘ We trained and tested models which employed the soft attention, GMM attention, and SAGMM-tr attention mechanisms as an encoder-decoder attention.’ in Section 4.2.
> >
> > **Comment** English usage. Just a few examples of grammar errors and unclear text, as there are too many to list.
> >
> > **Response** Thanks to the reviewer’s comments, we fixed the grammar errors in the paper. We are revising the paper and plan to update proofread in the rebuttal stage.
> >
> > **Comment** page 1, "mismatch between attention parameters from decoders and information distribution in encoder outputs". This sentence is difficult to parse. What is the mismatch here? Why would the source encoding and decoder query vector need to "match", especially in a purely location-based attention scheme?
> >
> > **Response** We rewrote confusing sentence and included figure 1 (c) to have a better understanding of GMM attention.
> >
> > **Comment** page 4: "fixed length with hyperparameters" What are they hyperparameters being referred to here?
> >
> > **Response** We add explanations and equations for SAGMM-tr with fixed attention window width as “The number of tokens we needs to wait …” in page 5, Sec. 2.4. $c$ refers to the attention window width in equation (17)-(20).
> >
> > **Comment** page 5, Sec. 4: "enables early inference". What does "early inference" mean?
> >
> > **Response** we corrected misleading term “early inference” to “online inference”.
> >
> > **Detailed comments**
> > > page 1, "attend subset of long sequences" is missing a preposition, e.g., "attend to a subset". It seems that "long sequences" is meant to refer a single source sequence.
> >
> > >page 5, Sec. 3: "for the inference" -> "for inference"
> >
> > >page 5, Sec 4.1: "1 second speeches" -> "1 second long utterances"
> >
> > >page 5, Sec 4.1: "from 30 vocabulary" -> "from a vocabulary of 30 words"
> >
> > **Response** According to reviewer’s comments, we fixed grammar error in this paper. We will check other grammar issues in the rest of rebuttal stage.
> >
> > **Comment** Sec 2.2: Sutskever et al., 2014 did not use content-based attention.
> >
> > **Response** According to the reviewer’s comment, we removed Sutskever et al., 2014 from the content-based attention reference.

---

### Official Review · AnonReviewer1 · 2020-10-29
**A new Gaussian-based attention mechanism by incorporating the source (key) information**

**Rating:** 6
**Confidence:** 5

**Review:**

This paper proposes a novel Gaussian mixture-based attention mechanism by incorporating the source (key) information, enabling a flexible representation of the Gaussian attention pattern with the well-described formulation. The effectiveness of the proposed method was validated by 1) artificially created long utterances, 2) Librispeech (segmented and concatenated utterances), and 3) machine translation. The paper also includes several attention visualizations to show the reasonable attention patterns obtained by the proposed method. The paper is well written. The precise monotonic attention gains a lot of attention for streaming ASR and simultaneous machine translation, and this paper would gein broad interests. My concerns about this paper are that 1) the novelty and the effectiveness are a little bit incremental, and 2) the paper requires more clarifications about the formulations and experimental descriptions (see my detailed comments).

Detailed comments:
- In the introduction, "The GMM attention is a pure location-aware algorithm in which the model selects attention windows cumulatively without considering source contents.": This is a little bit hard to understand. I understood it after I followed Section 2. It would be better to make this expression more understandable.
- I'm curious about the GMM's behavior (how the Gaussian parameters would be changed for each layer and each head). These are described in the appendix, but the authors may consider discussing it in the main paper. It is well known that the general soft attentions' behaviors are very different in the lower and higher layers. The lower-layer attentions are blur and less monotonic. I would like to ask the authors to discuss how the GMM parameters can adapt to capture such behaviors in the main paper.
- Equation (4): It's better to explain why the softplus function is used for \Delta and \Sigma. (monotonicity and positive constraint of the variance, right?)
- Equation (9): It's better to have some explanation about why $\delta$ is bounded by [0, 1].
- Equation (14): Is there proof?
- Section 2.4: Does the truncation make training faster?
- Section 4: The authors should clarify why they did not use a language model. Besides, it's better to clarify that the method does not use the language model in the table's caption (Is that correct? All the results listed in the table are without LMs, right?). The readers remember the Librispeech number with the LM, and they may confuse it.
- Why does the table 4 result not include the GMM attention result? It's better to compare the normal GMM and the proposed method in the online mode.

---

> ### Author Response · Authors · 2020-11-17
> **Response to Reviewer #1**
>
> We would like to thank the anonymous reviewers for their valuable feedback, and for the time and effort spent on evaluating our submission. The reviewers raised some interesting points, which we have carefully addressed, and as a result, we believe that the revised draft is significantly improved compared to the first submission.
>
> **Comment** In the introduction, "The GMM attention is a pure location-aware algorithm in which the model selects attention windows cumulatively without considering source contents.": This is a little bit hard to understand. I understood it after I followed Section 2. It would be better to make this expression more understandable.
>
> **Response** We rewrote the sentence for readability as “The GMM attention is a pure location-aware algorithm in which the encoder contents are not considered during attention score calculation.”
>
> **Comment** I'm curious about the GMM's behavior (how the Gaussian parameters would be changed for each layer and each head). These are described in the appendix, but the authors may consider discussing it in the main paper. It is well known that the general soft attentions' behaviors are very different in the lower and higher layers. The lower-layer attentions are blur and less monotonic. I would like to ask the authors to discuss how the GMM parameters can adapt to capture such behaviors in the main paper.
>
> **Response** We added the paragraph ‘Figure 3 shows...’ in the subsection 4.2. The lower layers of SAGMM-tr have their attentions more spread-out over a consecutive subsequence of the encoder tokens. The higher layers utilize the context vectors from the lower encoder-decoder attentions and relatively more focused, partly resembling the behavior of soft attentions.
>
> **Comment** Equation (4): It's better to explain why the softplus function is used for \Delta and \Sigma. (monotonicity and positive constraint of the variance, right?)
>
> **Response**
> Yes, we applied the softplus function to keep the monotonicity and positive constraints. The softplus function has been also motivated by the previous study in speech synthesis [Link](https://arxiv.org/abs/1910.10288).
>
> **Comment**  Equation (9): It's better to have some explanation about why $\delta$ is bounded by [0, 1].
>
> **Response** As shown in Figure 1, our goal is to extend the GMM attention by allowing variability in the interval lengths. Thus, the lower bound set to zero is natural. On the other hand, an interval too wide would harm the approximation of Gaussian distribution and we need some upper bound, and sigmoid is used widely to smoothly bound an arbitrary real number to fixed interval. Furthermore, CIF [Link](https://arxiv.org/abs/1905.11235) had similar idea for example.
>
> **Comment** Equation (14): Is there proof?
>
> **Response** Thanks to the reviewers’ comments, we removed Equation (14) from the original paper and revised the paragraph. The equation (14) was intended to show the similarity between attention score summation and integral of normal distribution. However, it was confusing and redundant in the paper.
>
> In the paragraph ‘The attention score for each encoder token…’ in page 4, we raised a point that since the GMM-based attentions are analogous to integral of normal distribution, the sum of attention score does not diverge without softmax operation over encoder tokens. Then, we can train model easily without numerical instability. The softmax-free property of SAGMM attention leads to online inference in SAGMM-tr by truncating long-tail of the normal distributions.
>
> **Comment** Section 2.4: Does the truncation make training faster?
>
> **Response** No, the truncation did not reduce the training time.
>
> **Comment** Section 4: The authors should clarify why they did not use a language model. Besides, it's better to clarify that the method does not use the language model in the table's caption (Is that correct? All the results listed in the table are without LMs, right?). The readers remember the Librispeech number with the LM, and they may confuse it.
>
> **Response** All results in our paper did not use an external language model (LM). We did not employ LM for easy comparison with several previous studies and our main interest is on-device applications in which the external LM is not popular yet. Though we cannot combine a language model in rebuttal stage, the performance with model with LM will be obtained in follow-up study.
>
>
> **Comment** Why does the table 4 result not include the GMM attention result? It's better to compare the normal GMM and the proposed method in the online mode.
>
> **Response**  As shown in Figure 1 and appendix, the truncated GMM model failed to learn the alignments and attended to future contexts during inference. We decided to not include the truncated GMM in table 4 since it could not run online inference in practice. The truncated GMM showed a WER of 4.03% in test-clean and 10.16% in test-other, respectively.

---

### Official Review · AnonReviewer4 · 2020-10-29
**Cool tweak to GMM attention, but very specific to ASR.**

**Rating:** 5
**Confidence:** 4

**Review:**

Summary:
This paper introduces “source-aware” GMM attention and applies it to offline, online, long-form ASR.  The value of source-aware GMM attention appears to be its ability to “ignore” long segments of silence in the input audio, which could potentially be more difficult to do using other attention mechanisms.  Fairly competitive results are presented for offline ASR.  For online ASR, the results are state-of-the-art amongst sequence-to-sequence-based models.

Reasons for score:
The main contribution of this paper seems to be the addition of predicted weights for each encoder step that allow the monotonic GMM attention mechanism to ignore certain input frames.  This seems to be directly useful in ASR; however, I’m not sure if it has any direct use outside of ASR.  The paper is also fairly hard to follow and not well motivated.  Because it is so ASR specific, I feel that it would be a better fit for a speech conference.


High-level Comments:
* Despite being a relatively minor tweak, the use of source-aware weights for each encoder step is a cool idea that integrates nicely into existing GMM-based attention mechanisms.
* As mentioned above, I felt that the paper was fairly hard to follow and not well motivated.  It took me until Figure 2 on page 6 to realize that the main benefit of using the source-aware mechanism was to ignore long segments of silence.  It would be good to provide this motivation earlier in the paper.


Detailed Comments:
* I can’t figure out why there are 4 attention plots in Figure 2 (it says it’s single-head).  It might be a good idea to clarify in the caption.
* I would recommend having your submission proof-read for English style and grammar issues.  The issues are subtle but addressing them would help to improve readability.
* Eq. (14) seems unnecessary (it follows from basic calculus and doesn’t apply in actual usage).  Is there a reason it was included?
* End of Section 2.2, “Moreover, the GMM attention score is uni-modal and cannot discard the non-informative tokens in attention window”: I'm not sure what is meant by "uni-modal" here.  A mixture of unimodal distributions (such as a GMM) is multi-modal.
* The addition of section 4.3 felt a bit tacked-on.  To me, it doesn’t seem worthwhile to attempt to apply a monotonic attention mechanism specifically designed for ASR to a non-monotonic task like MT.  If indeed there is an interesting contribution to be made via positive constraint relaxation, I would save all of Section 4.3 for the followup paper rather than inserting it at the end of this paper.

Update (2020-12-03):  Increasing score from 4 to 5.

---

> ### Author Response · Authors · 2020-11-17
> **Response to Reviewer #4**
>
> We would like to thank the anonymous reviewers for their valuable feedback, and for the time and effort spent on evaluating our submission. The reviewers raised some interesting points, which we have carefully addressed, and as a result, we believe that the revised draft is significantly improved compared to the first submission.
>
> **Comment** As mentioned above, I felt that the paper was fairly hard to follow and not well motivated. It took me until Figure 2 on page 6 to realize that the main benefit of using the source-aware mechanism was to ignore long segments of silence. It would be good to provide this motivation earlier in the paper.
>
> **Response** We added the attention plots from soft attention, truncated GMM attention, and SAGMM attention in figure 1. We also attached description on behavior of attention mechanisms in section 1.
>
> **Detailed Comments**
>
> **Comment** I can’t figure out why there are 4 attention plots in Figure 2 (it says it’s single-head). It might be a good idea to clarify in the caption.
>
> **Response** Each subplot in figure 3 (figure 2 in original paper) corresponds to the attention plot for each decoder layers. Four attention plots come from  4 decoder blocks by indicating the encoder-decoder attention.
>
> **Comment** I would recommend having your submission proof-read for English style and grammar issues. The issues are subtle but addressing them would help to improve readability.
>
> **Response** We are polishing up the paper to improve readability. We plan to update the proofread in the rebuttal stage.
>
> **Comment** Eq. (14) seems unnecessary (it follows from basic calculus and doesn’t apply in actual usage). Is there a reason it was included?
>
> **Response** Eq. (14) was included to emphasize the numerical stability of SAGMM algorithm. However, eq. (14) was confusing and seemed redundant in the paper. We removed Eq. (14) and revised the paragraph. In the paragraph ‘The attention score for each encoder token…’ in page 4, we raised a point that since the GMM-based attentions are analogous to integral of normal distribution, the sum of attention score does not diverge without softmax operation over encoder tokens. Then, we can train model easily without numerical instability. The softmax-free property of SAGMM attention leads to online inference in SAGMM-tr by truncating long-tail of the normal distributions.
>
> **Comment** End of Section 2.2, “Moreover, the GMM attention score is uni-modal and cannot discard the non-informative tokens in attention window”: I'm not sure what is meant by "uni-modal" here. A mixture of unimodal distributions (such as a GMM) is multi-modal.
>
> **Response** As the reviewer commented, GMM attention is multi-modal. Thanks to the reviewer’s comment, we have corrected the sentence as ‘Moreover, the number of modes in GMM attention is limited by the number of mixture components’.
>
> **Comment** The addition of section 4.3 felt a bit tacked-on. To me, it doesn’t seem worthwhile to attempt to apply a monotonic attention mechanism specifically designed for ASR to a non-monotonic task like MT. If indeed there is an interesting contribution to be made via positive constraint relaxation, I would save all of Section 4.3 for the followup paper rather than inserting it at the end of this paper.
>
> **Response** Previous studies on monotonic attention include experimental results in the non-monotonic natural language processing tasks. [Link](https://openreview.net/pdf?id=Hko85plCW) [Link](https://arxiv.org/abs/1906.05218) While our monotonic attention mechanism did not perform well with the non-monotonic tasks, we have experimented and attached the results for readers who would like to know the performance of the SAGMM-tr on non-monotonic tasks.

---

### Official Review · AnonReviewer5 · 2020-11-05
**Good paper if errors and confounding factors are resolved**

**Rating:** 5
**Confidence:** 4

**Review:**

This paper proposes a monotonic attention to improve the latency of decoding. The attention is an improvement over the attention based on Gaussian mixture model, allowing the attention weights to depend on the encoder outputs. The experimental numbers are strong.

I am leaning towards accepting the paper, but I do have a few concerns that the paper fails to address. Some of the concerns can be fixed, and some will need additional experiments. I will revise the score based on the authors' response.

The first concern is simply that there are errors and ambiguous statements in the paper. For example, equation (14) is obviously wrong, and in section 2.4 it is unclear how many frames the encoder needs to see before the decoder can predict the next token. The latter is crucial because streaming is the major reason behind the use of monotonic attention. A detailed list is given at the end.

The second concern is that many confounding factors are introduced in the experiments, presenting us from concluding that the improvement is solely coming from the use of SAGMM. For example, in the experiments, CTC is used as an additional loss to guide the learning of monotonic attention. During training, the paper also stochastically introduces an all one vector at the end of the input. It is unclear whether these two have an impact on learning monotonic attentions. It would be more convincing to have results without these additions, and would make the comparison to others more meaningful (assuming that others did not use the same additions).

One minor concern is that the learned representation depends on the attention used. The paper did mention this in the translation experiments, but it is under developed and no further hypotheses or evidence are provided. This would require almost a separate paper, but I believe this is key to understanding why regular content-based attention fails and whether we actually solves the problem.

Here are the detailed comments.

> ... the proposed attention mechanism solves the online and long-form speech recognition problems ...

This is a strong claim. I don't think the paper presents enough evidence that the problem is solved.

> equation (1), (2), and (3)

This is minor, but the variables soft_\alpha, head^h, and Multihead can be better typeset.

> equation (4)

This is again minor. \Sigma_i is a scalar, and it might be better to use the lower case \sigma_i.

> equation (13)

What are I and J?

> \mu_i = \mu_{i-1} + relu3(\Delta_i)

What is relu3?

> equation (14)

This equation is wrong in many ways. I assume j \to \infty means having infinitely many time steps, and \delta_j \to 0 applies to \delta_j for all j.

> 2.4 SAGMM-tr for online inference

In streaming mode, while the frames are coming in, how does the decoder know when to produce the next token?

> We tested the window width c = 15 centered by \mu_i ...

What is c?

> ... we randomly concatenate the 1-vector after the end of source sentence with probability p_{eos} ...

This trick is not implemented in other approaches. Have the authors tried without using this trick? How do know if the improvement is purely due to this trick and not about the proposed SAGMM?

> We adopt the CTC loss on the encoder output ...

The CTC loss is meant to help learn monotonic attention. Why bother using the CTC loss if the proposed attention is already monotonic? Does this suggest that the CTC loss has other added benefits?

> Streamable model, bi-directional enc. in Table 2

How are bidirectional encoders in the streaming case? Using a bidirectional encoder by definition cannot be used for streaming unless I missed something.

> Transformer (121M) (Ours) in Table 2

I assume this uses a regular content-based attention. Please clarify. I would even group the Transformer, GMM, and SAGMM-tr rows together, since they provide the control experiments to support the usefulness of the proposed method. The other rows are less useful, since the architectures are different. The comparison of absolute numbers tell us little, except that the numbers provided for the proposed approach are in the right ballpark.

---

> ### Author Response · Authors · 2020-11-17
> **Response to Reviewer #5 (1/2)**
>
> We would like to thank the anonymous reviewers for their valuable feedback, and for the time and effort spent on evaluating our submission. The reviewers raised some interesting points, which we have carefully addressed, and as a result, we believe that the revised draft is significantly improved compared to the first submission.
>
> **Comment** The first concern is simply that there are errors and ambiguous statements in the paper. For example, equation (14) is obviously wrong, and in section 2.4 it is unclear how many frames the encoder needs to see before the decoder can predict the next token. The latter is crucial because streaming is the major reason behind the use of monotonic attention. A detailed list is given at the end.
>
> **Response** We removed the equation (14) and added explanations about numerical stability. Since the GMM-based attentions are analogous to integral of normal distribution, the sum of attention score does not diverge without softmax operation. Then, we can train easily model without numerical instability. The softmax-free property of SAGMM attention leads to online inference in SAGMM-tr.
>
> We added equations (16), (20) to show the number of encoder frames needed to predict the attention output. In the adaptive window model, the number of frames decoder waits depends on $\sigma_{i}$. In the fixed window model, the model needs to wait $\frac{c}{2}$ frames after $\nu_{j}> \mu_{i}$ is satisfied. In experiments, the model waits 7 more frames with $c=15$.
>
> **Comment**  The second concern is that many confounding factors are introduced in the experiments, presenting us from concluding that the improvement is solely coming from the use of SAGMM. For example, in the experiments, CTC is used as an additional loss to guide the learning of monotonic attention. During training, the paper also stochastically introduces an all one vector at the end of the input. It is unclear whether these two have an impact on learning monotonic attentions. It would be more convincing to have results without these additions, and would make the comparison to others more meaningful (assuming that others did not use the same additions).
>
> **Response** In this paper, we adopt the CTC loss as an auxiliary loss for fair comparison with previous studies. The model without CTC loss learned similar mappings with that with CTC loss and the WER of the test-clean/test-other set was 3.84%/9.86%, respectively. Therefore, the model still learns monotonic mapping without an auxiliary loss.
>
> The 1-vector trick has been developed for our in-house datasets in which EOS is allowed only when the user closes session. In common environments like LibriSpeech, the 1-vector trick does not have crucial role. Due to lack of time, we could not train the model without 1-vector trick from scratch. Instead, we fine-tuned existing SAGMM-tr model. The results is given as:
>
> (w/ 1-vector trick) test-clean: 3.52%  test-other: 9.29% WER
>
> (w/o 1-vector trick) test-clean: 3.54%, test-other: 9.37% WER
>
> From the result, the 1-vector trick is not significant to SAGMM-tr in general environments.
>
> **Comment**  One minor concern is that the learned representation depends on the attention used. The paper did mention this in the translation experiments, but it is under developed and no further hypotheses or evidence are provided. This would require almost a separate paper, but I believe this is key to understanding why regular content-based attention fails and whether we actually solves the problem.
>
>  **Response** We suppose that the failure of the content-based attention partly comes from the softmax operation over all encoder outputs, which makes the model unstable when the similar contents from the encoder contents appears multiple times.
>  In SAGMM-tr, the attention scores follows truncated normal distribution that the model attends to a subset of sequences.  Discarding tokens in irrelevant positions would be helpful to learn stable alignments. We did not include this discussion in the main paper since this assumption should be verified via a variety of tasks in following works.

---

> > ### Author Response · Authors · 2020-11-17
> > **Response to Reviewer #5 (2/2)**
> >
> > **Responses for detailed Comment**
> >
> > **Comment** ... the proposed attention mechanism solves the online and long-form speech recognition problems ...
> > This is a strong claim. I don't think the paper presents enough evidence that the problem is solved.
> >
> > **Response** Thanks to the reviewer's comment, we revised the sentence as ‘…the proposed attention mechanism improves the performance on the online and long-form speech recognition problems…’
> >
> > **Comment** equation (1), (2), and (3)
> >
> >  This is minor, but the variables soft_\alpha, head^h, and Multihead can be better typeset.
> >
> > **Response** We revised typeset as : $head^{h}$ to $H^{h}$, $Multihead$ to  $M$, $Soft$ _ ${\alpha}, GMM{\alpha}, ...$ to $\alpha_{Soft}, \alpha_{GMM}, ... $
> >
> > **Comment** equation (4)
> >
> >  This is again minor. \Sigma_i is a scalar, and it might be better to use the lower case \sigma_i.
> >
> > **Response** Thanks to the reviewer’s comment, changed $\Sigma$ to $\sigma$ for better readability.
> >
> > **Comment** equation (13)
> >
> >  What are I and J?
> >
> > **Response** We add explanation for I and J as ‘where I and J denote the length of decoder and encoder sequences, respectively’.
> >
> > **Comment**\mu_i = \mu_{i-1} + relu3(\Delta_i)
> >
> > What is relu3?
> >
> > **Response** We clarify relu3 activation function as min( max($\Delta_i$, 0), 3).
> >
> > **Comment** equation (14)
> >
> > This equation is wrong in many ways. I assume j \to \infty means having infinitely many time steps, and \delta_j \to 0 applies to \delta_j for all j.
> >
> > **Response** As response to the above comment, we removed (14) and revised confusing sentences.  The response to above comment also summarizes the revised paragraph.
> >
> > **Comment**  In streaming mode, while the frames are coming in, how does the decoder know when to produce the next token?
> >  We tested the window width c = 15 centered by \mu_i ...
> >
> > What is c?
> >
> > **Response** We revised the section 2.4 and clarified when the condition to predict the next token. In the adaptive window algorithm, the model needs to wait until $\nu_{j}$ becomes larger than $\mu_{i}  + 2\sqrt{\sigma_{i}}$. In the fixed window algorithm, the model waits $\frac{c}{2}$ frames after $\nu_{j}>\mu_{i}$.
> >
> > $c$ is attention window size which leads to maximum latency of the model in equation (20).
> >
> > **Comment** ... we randomly concatenate the 1-vector after the end of source sentence with probability p_{eos} ...
> >  This trick is not implemented in other approaches. Have the authors tried without using this trick? How do know if the improvement is purely due to this trick and not about the proposed SAGMM?
> >
> > **Response** As described in the above comment, the 1-vector trick had minor improvements on SAGMM models.
> >
> > **Comment** We adopt the CTC loss on the encoder output ...
> >
> > The CTC loss is meant to help learn monotonic attention. Why bother using the CTC loss if the proposed attention is already monotonic? Does this suggest that the CTC loss has other added benefits?
> >
> > **Response** The CTC loss is adopted for fair comparison with previous studies. Without auxiliary CTC loss, the performance of model dropped 0.32% in test-clean set. CTC loss calculates marginalization over all possible alignments, and it may help to emphasize the "correct" sequences in the intermediate layers.
> >
> > **Comment** Streamable model, bi-directional enc. in Table 2
> >
> > How are bidirectional encoders in the streaming case? Using a bidirectional encoder by definition cannot be used for streaming unless I missed something.
> >
> > **Response** We clarified the notation as ‘Streamable decoders, bi-directional encoders’. All models in table 2 are offline models.
> >
> > **Comment** Transformer (121M) (Ours) in Table 2
> >
> > I assume this uses a regular content-based attention. Please clarify. I would even group the Transformer, GMM, and SAGMM-tr rows together, since they provide the control experiments to support the usefulness of the proposed method. The other rows are less useful, since the architectures are different. The comparison of absolute numbers tell us little, except that the numbers provided for the proposed approach are in the right ballpark.
> >
> > **Response** We clarified the notation as Soft attention, GMM, and SAGMM-tr for readability. The difference between three models is encoder-decoder attention scheme. We also separate our models from previous works for better comparison.

---

### Author Response · Authors · 2020-11-23
**Upload proofread**

We revised grammar errors on our paper for better readability. We would like to thank the anonymous reviewers for their valuable feedback.

---

### Decision · Program_Chairs · 2021-01-07
**Final Decision**

**Decision:**

Reject

**Comment:**

The paper proposed a useful incremental extension to the monotonic GMM attention by incorporating source content.
It has shown comparable performance for online and long-form speech recognition, but falls behind on the machine translation task. For online ASR, it would be more convincing to include latency comparisons across different streaming models besides WERs.
The presentation of the paper can be further improved although it already got better based on reviewers' comments.
As in the discussion, a more accurate description of the method would be "multi-head Gaussian attention" instead of GMM attention.

The main factor for the decision is limited novelty and clarity can be further improved.